# Genetic rescue increases fitness and aids rapid recovery of an endangered marsupial population

Andrew R. Weeks [1,2], Dean Heinze[3], Louise Perrin[4], Jakub Stoklosa[5], Ary A. Hoffmann[1], Anthony van Rooyen[2], Tom Kelly[2,4] & Ian Mansergh[3]

Genetic rescue has now been attempted in several threatened species, but the contribution of genetics per se to any increase in population health can be hard to identify. Rescue is expected to be particularly useful when individuals are introduced into small isolated populations with low levels of genetic variation. Here we consider such a situation by documenting genetic rescue in the mountain pygmy possum, *Burramys parvus*. Rapid population recovery occurred in the target population after the introduction of a small number of males from a large genetically diverged population. Initial hybrid fitness was more than two-fold higher than non-hybrids; hybrid animals had a larger body size, and female hybrids produced more pouch young and lived longer. Genetic rescue likely contributed to the largest population size ever being recorded at this site. These data point to genetic rescue as being a potentially useful option for the recovery of small threatened populations.

[1] School of BioSciences, Bio21 Institute, The University of Melbourne, 30 Flemington Road, Parkville, VIC 3010, Australia. [2] cesar Pty Ltd, 293 Royal Parade, Parkville, VIC 3052, Australia. [3] Research Centre of Applied Alpine Ecology, La Trobe University, Melbourne, 3086 VIC, Australia. [4] Mt Buller Mt Stirling Resort Management, Mt Buller, VIC 3723, Australia. [5] School of Mathematics and Statistics and Evolution and Ecology Research Centre, The University of New South Wales, Sydney, NSW 2052, Australia. Correspondence and requests for materials should be addressed to A.R.W. (email: aweeks@unimelb.edu.au)

There is increasing interest in using genetic translocations as a means of recovering small populations of threatened species, particularly with the growing urgency of needing to maintain evolutionary adaptive capacity and reduce inbreeding effects under rapidly changing environments[1–6]. Many threatened species now include small and fragmented populations that may have been isolated for long periods of time[7]. In the past, conservation biologists have avoided crossing genetically separate populations because of the perceived risk of outbreeding depression[8, 9]. However, this risk is likely to have been overstated[10] and there is potential for assisted gene flow to increase population fitness and adaptability[1, 7]. Nevertheless, to date, there are very few examples where the benefits of genetic translocations have been documented and the fitness effects of population hybrids and derived genotypes measured[1]. Similarly, the few rare examples are often accompanied by environmental improvements, making it difficult to separate genetic from environmental effects[9].

*Burramys parvus* is one of Australia's most threatened marsupials. It is the only hibernating marsupial, and is restricted to the alpine regions of Australia. There are three main regions where *B. parvus* is found in similar alpine habitats (Supplementary Fig. 1) and populations in these regions are genetically separate, having been isolated for at least 20,000 years[11]. The southern population is restricted to Mount Buller and contained entirely within the Mount Buller Alpine Resort (Supplementary Fig. 2). This population is regarded as highly threatened, having undergone a rapid decline in genetic diversity that paralleled a demographic collapse[12]. Because the population was predicted to become extinct, a recovery programme was implemented that involved habitat restoration, predator control and environmental protection, along with the introduction of males from healthy and genetically variable populations of *B. parvus* from the centre of its distribution in 2011 and 2014. Restoration was used to link favourable patches by creating boulderfields (prime habitat for females) and through revegetation (secondary habitat for males) (Supplementary Fig. 2). These measures were associated with a rapid increase in population size of *B. parvus* in the main habitat area at Mount Buller, the Federation-Wombat bowl, and we show that genetic rescue likely contributed to this increase.

## Results

### Population size at Mount Buller.
Population size for the Federation-Wombat bowl was estimated based on the capture–recapture data from annual spring monitoring (see "Methods" section), where 70-96% of the adult population is captured in most years. A rapid increase in population size occurred between 2008 and 2015; in fact, the adult population is now 68% larger than when this population was first discovered in 1996 (Fig. 1). Prior to the translocations, the population increased by ~40 individuals (Fig. 1) across three sampling periods (2008–2011) likely as a consequence of habitat improvement and predator control programs. After the translocations (2012 onwards), there was a more rapid increase of around 100 individuals across a similar time interval. This provides an opportunity to investigate the potential importance of genetic factors in the recovery process. Due to the release of some individuals from an independent programme in spring 2013 (see "Methods" section), we focus our analyses on the 2011–2013 data and individuals trapped during this period.

### Contribution of introduced males and genetic diversity.
The genetic material that was first introduced into the population consisted of five males from Mount Higginbotham (Supplementary Fig. 1) in 2011, with four contributing to F1s during the 2011

breeding season. These four males only survived the 2011 breeding season, while the fifth male survived until 2015, but did not start contributing to offspring until the 2013 breeding season. In spring 2014, six males were then introduced to the Mount Buller population from the Timms Spur population in the central region during the breeding season, with four of the males contributing during that breeding season. Alleles from introduced males have now become integrated into the gene pool (Supplementary Fig. 3) and levels of heterozygosity are now approaching healthy populations of *B. parvus* in the central region (Table 1). Genetic diversity (allelic richness, heterozygosity) has increased along with population size as a consequence of these introductions (Table 1).

### Fitness of hybrids.
The F1s from the initial introduction in 2011 contributed genetic material to adults that were collected in 2012 during annual spring monitoring and had not been recorded before this time. As there were estimated to be 21 resident males present in the population in 2011 (see "Methods" section), we calculated the fitness of the introduced males compared to resident males using the frequency of hybrid genotypes in 2012. These point to a fitness advantage to the introduced males and/or greater F1 hybrid juvenile survival with the total fitness advantage being greater than twice that of the local males/juveniles

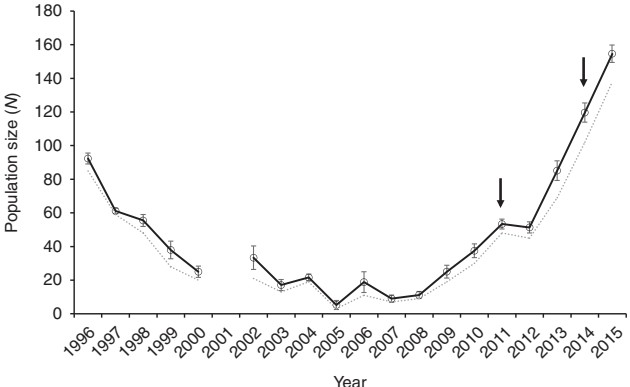

**Fig. 1** *Burramys parvus* adult population size. Estimates are for the Federation-Wombat bowl area at Mount Buller based on the capture-recapture data. Solid line is the estimate based on the robust design model[22] with standard error (mean) bars. Dashed line represents the number of unique observed individuals. Arrows indicate the 2011 and 2014 introduction of six males from each of Mount Higginbotham and Timms Spur in the central region (Supplementary Fig. 1)

**Table 1 Temporal changes in genetic diversity within the Mount Buller population**

| Population | Year | $n$ | $N_a$ | $A_r$ | $H_O$ | $H_E$ | $F_{IS}$ |
|---|---|---|---|---|---|---|---|
| Mt Buller | 2010 | 29 | 1.583 | 1.515 | 0.141 | 0.139 | −0.014 |
| Mt Buller | 2011 | 40 | 1.708 | 1.594 | 0.160 | 0.164 | 0.024 |
| Mt Buller | 2012 | 43 | 3.625 | 3.107 | 0.333 | 0.309 | −0.077 |
| Mt Buller | 2013 | 67 | 3.750 | 3.135 | 0.329 | 0.327 | −0.008 |
| Mt Buller | 2014 | 103 | 3.875 | 3.287 | 0.352 | 0.355 | 0.006 |
| Mt Buller | 2015 | 138 | 4.583 | 3.608 | 0.392 | 0.416 | 0.057 |
| Mt Higginbotham | 2012 | 104 | 5.583 | 4.524 | 0.526 | 0.552 | 0.047 |

Population genetic statistics for *B. parvus* based on 24 nuclear microsatellite markers. Sample size ($n$), average number of alleles ($N_a$), mean allelic richness ($A_r$), mean observed ($H_O$) and expected ($H_E$) heterozygosity, and the inbreeding coefficient ($F_{IS}$). Estimates for a large and stable population in the central region (Mount Higginbotham) are shown as a comparison

| Table 2 Number of new adult hybrids, non-hybrids and their relative fitness | | | | |
|---|---|---|---|---|
| Year | Hybrids | Non-hybrids | Hybrid Relative Fitness (bootstrap CI) | Significance |
| 2012 | 13 (6.54) | 21 (27.46) | 2.60 (1.29–4.72) | P = 0.005 |
| 2013 | 29 (23.23) | 18 (23.77) | 1.65 (0.94–2.98) | P = 0.092 |

Observed (expected, see "Methods" section) first year adult hybrids and non-hybrids in the population in 2012 and 2013, hybrid relative fitness compared with non-hybrids, and the significance of the difference based on a one-tailed Fisher's exact test

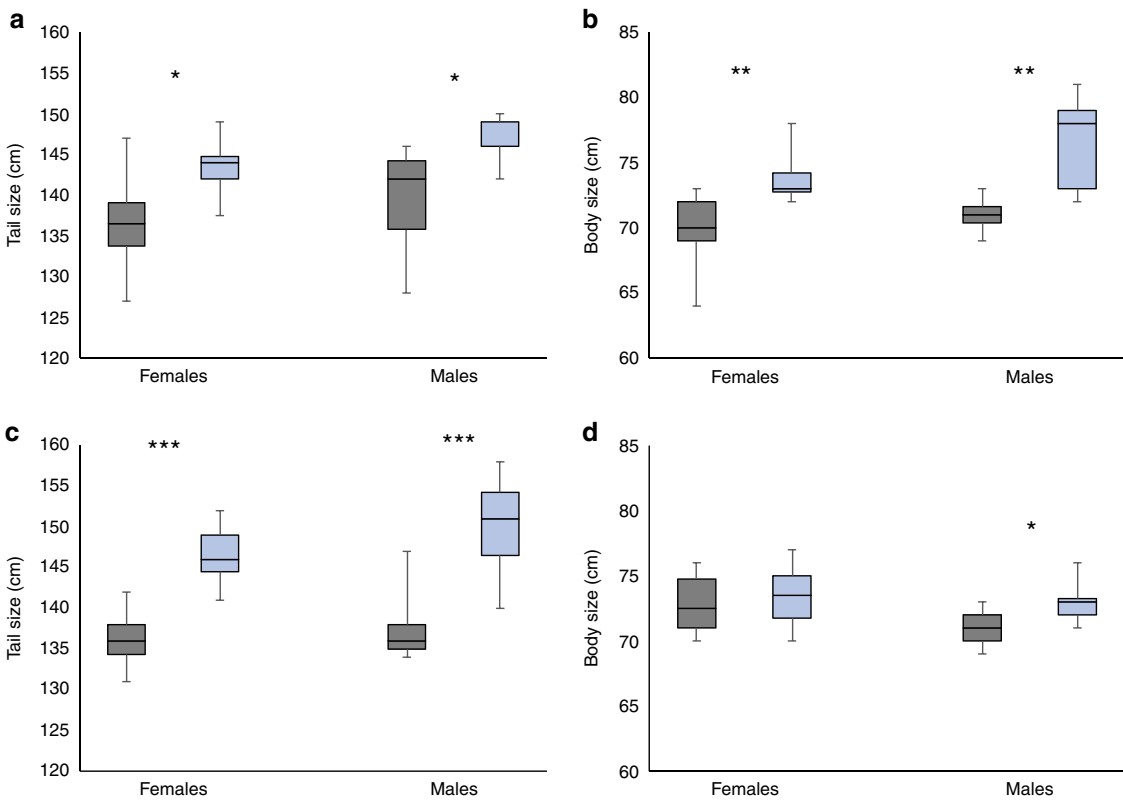

**Fig. 2** Size differences between hybrid and non-hybrid females and males. Box plots of tail and body size for new adults in 2012 (**a**, **b**) and 2013 (**c**, **d**). The rectangle spans the first to third quartile, the segment inside the rectangle represents the median, and the whiskers above and below show the maximum and minimum value. Grey boxes are non-hybrids and light-blue boxes are hybrids. Significant differences, as assessed by ANOVA, between hybrids and non-hybrids are indicated (*P < 0.05, **P < 0.01, ***P < 0.001)

(Table 2). This heterotic effect may reflect fitness consequences of inbreeding in the Mount Buller population; there was a 76% drop in heterozygosity between 1996 and 2010, which equates to an effective population size of 3.88 (95% CIs; 1.81–8.05) during this period and led to detectable inbreeding (see "Methods" section). F1 males with hybrid genotypes were also larger than resident individuals, a pattern that was evident in both sexes (Fig. 2). All F1 hybrid females had a full-complement (four) of pouch young, whereas many non-hybrid females had less than four pouch young, and this reduction was marginally significant by a Fisher's exact test (P = 0.047). These data point to a substantial initial benefit from the genetic rescue, which was expected if the effects of deleterious alleles present in the Mount Buller population are masked in hybrids following population hybridisation.

We also considered the longevity of the F1 hybrids in the Mount Buller population compared to those without hybrid alleles from the same cohort up to spring 2015. For the males, 13 out of 14 survived only 1 year, regardless of whether they were hybrids. Females generally survived longer than males, with 15 out of 24 (62.5%) surviving into the second year, and the mean longevity for hybrids (2.78 years) was longer than for non-hybrids

(1.8 years). Four of the eight F1 hybrid females were still alive in spring 2015, whereas none of the 16 F1 non-hybrid females were known to be alive (Fisher's exact test, P = 0.008).

Adults sampled for the first time in 2013 were again genotyped and used to estimate fitness of resident (non-hybrid) vs. hybrid individuals. This comparison is complicated by the fact that there are now F2 hybrids and backcross individuals in the population as well as surviving F1 hybrids. We estimated the expected number of individuals with introduced alleles based on the total number of hybrids/non-hybrids in the population in 2012 (Methods). A total of 47 new adults were captured in spring 2013, which resulted in a 66% increase in population size in the Federation-Wombat bowl (Fig. 1). Of the 47 new adults, 29 carried introduced alleles with 6 identified as F2 hybrids and 23 identified as backcross individuals (no new F1 hybrids were detected). This increase in hybrid individuals is marginally non-significant (P = 0.09), but represents a hybrid fitness advantage of 1.65 relative to the resident (non-hybrid) adults (Table 2). As in 2012, individuals carrying hybrid alleles tended to be larger than those that did not, particularly for the males (Fig. 2), likely suggesting a relatively higher fitness.

**No evidence of outbreeding depression.** On the basis of the frequency of hybrids in 2012 (27.6% of females, 31.3% of males) and assuming random mating, we expected 8.6% of the new adults to represent F2s, and 41.6% of the population to represent backcrosses. The incidence of F2s (12.8%) and backcrossed individuals (48.9%) observed in the population were non-significantly higher than those expected. Mating ability of F1s and survival of the F2s, and backcrosses therefore appears to be normal (under the assumption of random mating). Similarly, there is no difference in the size of F2s and backcrossed females (tail $F_{1,16} = 1.292$, $P = 0.275$; body $F_{1,16} = 0.506$, $P = 0.489$; head $F_{1,16} = 0.018$, $P = 0.896$), or when both are compared to the surviving F1 hybrid females (tail $F_{1,21} = 1.025$, $P = 0.324$; body $F_{1,21} = 1.472$, $P = 0.240$; head $F_{1,21} = 0.617$, $P = 0.442$). While this does not provide a direct test of outbreeding depression, it suggests that under the assumption of random mating outbreeding depression effects are likely to be relatively small or non-existent.

## Discussion

Although it is difficult to separate the effects of genetic rescue from the effects of environmental improvements, the data suggest that genetic rescue has probably contributed directly to the increase in population size of *B. parvus* within the Federation-Wombat bowl area at Mount Buller. While environmental improvements likely led to a small initial increase in population size at Mount Buller, the further rapid increase in size is likely to have occurred because of genetic improvements. This is based on the direct fitness estimates that we have made in the population, and is also supported by the fact that similar environmental improvement programs for *B. parvus* in the central region have not resulted in population increases as seen at Mount Buller (Supplementary Fig. 4). Although it is hard to tease apart the direct contributions of environmental and genetic factors to population improvement, our results suggest that genetic rescue further increased population size after threatening processes had been mitigated.

The translocation of six males in 2014 from the Timms Spur population to the Mount Buller population further elevated genetic diversity by 15–20% (Table 1), and had the aim of genetic restoration[13] and increasing population resilience[1, 6]. However, the Mount Buller population still remains relatively small and population growth needs to continue to reduce extinction risk[14]. Further linking of habitat areas on Mount Buller to the Federation-Wombat bowl (Supplementary Fig. 2, regions B and C) and/or intra-site translocations are likely to be critical for population expansion and ensuring the long-term persistence of the Mount Buller population.

Our data show that, despite isolation of 20,000 years or more[11] of the populations used in this genetic rescue, there appear to be clear benefits associated with the introduction of genetic material from the central region into the Mount Buller population. Despite the long period of isolation between the Mount Buller and Mount Higginbotham populations, the alpine environments in these locations are similar in consisting of alpine herbfield, sedgeland and grassland interspersed by areas of heathland, and intersected by basalt-granite boulderfields. Fixed chromosomal differences are unlikely between the two populations as all pygmy possums share the same ancestral karyotype[15]. Genetic differences between populations are likely to be largely due to drift[7], and therefore we did not expect to see any evidence for outbreeding depression based on the decision tree in Frankham et al.[10].

A large number of other species could potentially benefit from genetic rescue because many threatened species exist as small isolated populations lacking genetic variation alongside larger populations with higher levels of genetic variation[7]. Genetic rescue can also be beneficial when only inbred populations are available as sources[3, 16, 17], although the benefits are likely to be less than from crosses with an outbred population[3] and there is a risk of an increasing genetic (drift) load. Subsequent population expansion is crucial for the lasting impacts of genetic rescue to be realised; otherwise deleterious alleles might only be masked for a generation and then potentially accumulate[14]. Ecological restoration programs to increase population size are therefore essential to the long-term success of any attempts to use genetic rescue[14]. An increase in adaptive responses is likely as a consequence of increased genetic variation[6, 7, 18–20]. We also emphasise that ongoing monitoring of the focal population is needed to adequately quantify the fitness benefits associated with the genetic changes. Future efforts at Mount Buller could be focused on genomic analyses of hybrid gene pools, which will provide insight into whether fitness effects of genetic rescue are associated with particular genomic regions and the long-term persistence of introduced vs. resident genes.

## Methods

**Sites.** The Mount Buller Alpine Resort (−37.146229° N, 146.441285° E) is located ~150 km north-east of Melbourne (Australia) and extends over ~850 ha between 1400 and 1800 m elevation (Supplementary Fig. 1). The entire habitat of *B. parvus* is contained within the resort boundaries (known as the southern region of the *B. parvus* distribution[11]), with the Federation-Wombat bowl the most significant area of *B. parvus* breeding habitat comprised of deep peri-glacial boulderfields and dense heath of mountain plum pine (*Podocarpus lawrenceii*) (Supplementary Fig. 2). Mount Higginbotham (−36.986461° N, 147.144569° E) is found within the Mount Hotham Alpine Resort, approximately 65 km north-east of Mount Buller, in the central region of the *B. parvus* distribution[11]. Timms Spur (−36.794306° N, 147.275346° E) is within the Alpine National Park, ~25 km north-east of Mt Higginbotham, and is the most northern site where *B. parvus* are known to occur within the central region.

**Trapping.** Live trapping of *B. parvus* was undertaken annually in spring at Mount Buller from 1996, with the exception of spring 2001, where no trapping was undertaken. Live trapping occurred at approximately the same time each year (between last week of October and first two weeks of November), with the variation in dates dependent on spring snow melt and approximate timing of mating. Trapping was confined to this period to capture females when they are carrying pouch young. Exceptions to this were years 1996–1998, where trapping occurred between October and December. Standard trapping methods[21] were followed and undertaken with Elliott type-A live-capture traps (Elliott Scientific, Upwey, Victoria, Australia). Traps were baited with walnuts, wood wool was placed in traps for bedding material, and plastic bags wrapped around the outside for insulation from inclement weather.

Each annual trapping event at Mount Buller consisted of setting Elliott traps at the same sites in the same grid pattern in regions A (Federation-Wombat bowl), B (Fanny's Finish / Summit South) and C (Grimus) (Supplementary Fig. 2). Region D (Summit North) was left as a reference site and has only been trapped twice, in spring 2004 and 2013. A total of 2010 trap nights were undertaken each annual trapping event, with 1275 undertaken in the Federation-Wombat bowl (region A). First time *B. parvus* captures were ear tagged (National Band & Tag Company, Newport, Kentucky, USA) with a unique number, measured for several morphometric / meristic traits (weight, tail/head/body length, number of pouch young (female), testes size (males), reproductive condition (females and males)), and had a hair or tissue (2 mm ear biopsy) sample taken (stored in > 95% ethanol). For recaptures, ear tag number was recorded and weight, number of pouch young, testes size and reproductive condition were remeasured.

**Translocations.** A translocation of six male *B. parvus* from Mount Higginbotham (central region) to Mount Buller was attempted in spring 2010, but due to delayed permission by government authorities, the translocation occurred too late in the breeding season (mid-October) for success, with males removed in November. The translocation was repeated in mid-September 2011 during the onset of snow melt; six *B. parvus* adult males were trapped (using Elliott traps as above) at Mt Higginbotham over a single night, translocated to Mount Buller the next day, and released in the Federation boulderfield (region A; Supplementary Fig. 2). Males were chosen for translocation for several reasons: (i) males were likely to be limiting in the Mount Buller population, (ii) males have the opportunity to mate with multiple females, and therefore a greater chance to produce multiple hybrid litters, (iii) males only live for one season on average, and (iv) the use of males maintained the maternal lineage on Mount Buller, which was a State Government of Victoria requirement. All released males were radio-tracked for three weeks after release to determine survival during this period. One male was not detected soon after release

(day 3 post release) and was an assumed mortality. The other five translocated males were detected throughout the 3 weeks, and all five were trapped during spring monitoring in early November confirming their survival throughout the breeding period. Only one male was known to survive through to the next breeding season (spring 2012).

Six females and seven males from a failed captive breeding programme at Healesville Sanctuary (Healesville, Victoria) were translocated in spring-summer 2013. This release was government approved, but independent of the in situ genetic rescue and on ground environmental works. These individuals had either a Mount Buller genetic background (two males) or were derived from a single cross between a female from Mount Buller and a male from Mount Higginbotham. Females were translocated in mid-October, while males were translocated in late November (to avoid competition with other males during breeding). Monitoring in January and February indicated that the translocated males were still sexually active (large testicles), while all other males in the population were not sexually active (contracted testicles). The translocated males were then removed in early March, due to fears that they could cause late litters, placing females at risk prior to hibernation. Only one of the translocated females was known to survive through to the next breeding season (captured during annual trapping in early November 2014).

We implemented another translocation of central male *B. parvus* to Mount Buller in spring 2014 to further increase genetic diversity and undertake genetic restoration[6, 13]. Six males were captured overnight (as above) on Timms Spur (Supplementary Fig. 1) in late September 2014 and released at Mount Buller on the same day in the evening within the Federation-Wombat bowl region. These males were not radio-tracked; however, five were trapped during annual spring monitoring in 2014, confirming their presence during the main breeding period. All translocations and trapping events were authorised and in accordance with ethical standards of relevant authorities in Victoria, Australia.

**Population size estimates.** All adult spring trapping data from the Federation-Wombat bowl sites (region A; Supplementary Fig. 2) undertaken since 1996 was used to construct individual capture histories; that is, an individual's capture (1) or non-capture (0). No translocated individuals were included in the analyses (removed from individual capture histories). There were 19 primary capture occasions (spanning the 20 years) with the number of secondary capture occasions varying from 5 to 20 (that is, the number of trapping nights within each primary occasion). We assume births and deaths, as well as emigration and immigration, and therefore analysed the data using open population capture–recapture models. Specifically, we used the robust design model[22], which assumes an open population across the primary occasions (that is, across each year) and a closed population (that is, constant births and deaths, emigration and immigration) across the secondary occasions within each primary occasion.

Various robust design models were fitted depending on how capture and survival probabilities were parametrised. To account for temporal effects, both capture and survival probabilities can be either time dependent or constant across capture occasions. Heterogeneity amongst individuals is modelled using a gender covariate, that is, we model both capture and survival probabilities as functions of a sex covariate. Any unexplained heterogeneity in capture probabilities is modelled through latent variables via a finite mixture-Huggins model[23]. For these mixture models, any individual that is part of the population is assumed to belong to a class $j$ with probability $\pi_j$. In our analysis, we use the default setting of $j = 2$ mixture classes, with mixture probabilities set to $\pi_1$ and $\pi_2$. Finally, to account for possible temporary emigration in the population, we consider several temporary emigration structures, these can be either random or Markovian types[22] (see also Supplementary Table 1 for further details). The robust design model likelihood is reparametrised according to the specified temporary emigration type[22]. We also fit models that excluded temporary emigration altogether.

We used the R-package RMark (an R language version of program MARK[24]) to fit models via maximum likelihood and the Bayesian information criterion to select the final model among a set of candidate models (see Supplementary Table 1 for a description of all fitted models). The smallest Bayesian information criterion was reported for Model 8 (see Supplementary Table 1). This final selected model is described as follows: survival probabilities are constant across primary occasions and dependent on gender; capture probabilities vary across primary occasions and are dependent on gender; mixture probabilities vary across primary occasions; and temporary emigration is of Markovian type and constant across primary occasions. Annual population size estimates (and standard errors) are then acquired from Model 8 using RMark.

To calculate the expected number of hybrids in 2012 (see below), we first obtained estimates of male population size for 2011. Due to the sparseness of the male capture–recapture data, we encountered several boundary estimates when fitting the robust design model on male-only data. To avoid computational instability and enhance precision in parameter estimates, we used specific closed population models. We fitted continuous-time capture–recapture models[25] via the software CARE-3. These models are more efficient than discrete-time capture–recapture models when the number of capture occasions are small[26]. Maximum likelihood was once again used for estimation, and male population size estimates (and standard errors) were obtained from CARE-3.

**Genetic and phenotypic analyses.** All *B. parvus* samples (hair and tissue) collected from the Mount Buller population from 2010–2015 and all translocated individuals were genotyped at 24 microsatellite loci. We genotyped samples with eight previously isolated microsatellite markers[11] and generated another 16 new markers using an established approach[27]. Briefly, the 454 sequencing platform was used to characterise microsatellite markers from 10 µg of genomic DNA extracted (using a Qiagen DNA Easy Kit, Qiagen) from ear tissue from a single *B. parvus* adult female collected from Mount Buller in 2010. The DNA was nebulised, ligated with 454 sequencing primers and subjected to high throughput DNA sequencing using the Roche GS FLX (454) system at the Australian Genome Research Facility (AGRF, Brisbane, Australia). The software QDD[28] and PRIMER 3[29] were used to select unique sequence contigs and design primer sets. Forty potential microsatellite loci were screened for polymorphism using eight template DNAs, representing samples from Mount Buller (four individuals) and Mount Higginbotham (four individuals). Loci were pooled into ten groups of four, labelled with unique fluorophores (FAM, NED, VIC, PET) and coamplified by multiplex PCR using a Qiagen multiplex kit (Qiagen) and an Eppendorf Mastercycler *S* gradient PCR machine[30]. Genotyping was subsequently performed using an Applied Biosystems 3730 capillary analyser (AGRF, Melbourne, Australia) and product lengths were scored manually and assessed for polymorphism using GeneMapper version 4.0 (Applied Biosystems). Sixteen microsatellite markers were selected (Genbank Accession numbers MF568684-MF568699), with some of these new markers chosen because they showed fixed differences or non-overlapping alleles between the Mount Buller and central region populations, and therefore were useful for tracking hybrid individuals. No randomisation or blinding was used for groups, although all phenotypic measures were recorded prior to genetic analyses revealed an individual to be a hybrid / non-hybrid.

Yearly population estimates of genetic diversity were calculated using GenAlEx version 6.5[31] ($N_a$, $H_O$, $H_E$) and FSTAT[32] ($A_r$, $F_{IS}$). Unique alleles across the 24 microsatellites were used to identify hybrid individuals that were derived from the 2011 translocated males in the spring 2012 (F1 hybrids) and 2013 populations. STRUCTURE[33] was also used to assess the genetic background of individuals (including the translocated males from 2011 and 2014). In the STRUCTURE analysis, we set $K = 2$ (representing the central and southern backgrounds) and took the average membership coefficient of 10 independent simulations (admixture model, allele frequencies independent, burn-in = 100,000, data iterations = 1,000,000; the admixture model was chosen to identify hybrids and the independent allele frequencies model was chosen because of the two very different genetic backgrounds[11, 33]). The effective population size between 1996 and 2010 was estimated from the change in expected heterozygosity[34] for the 8 microsatellite loci genotyped in[12] using the formula $H_t = H_0 \left(1 - \frac{1}{2Ne}\right)^t$, assuming a generation time of 1.54 years (based on survival probability estimated from the capture–recapture data between 1996 and 2010 and the average number of pouch young for different age classes of females[35]). Confidence intervals were calculated from the raw data.

We focused our analyses on the 2011–2013 spring data sets due to the confounding effects of the captive-bred individuals released in spring 2013 affecting the results in 2014 and 2015. The 2013 data set was not affected by released captive-bred males because they were released after breeding/spring trapping. In spring 2012, there were 8 female and 5 male F1 hybrids derived from four of the translocated Mount Higginbotham males compared with 21 females and 11 males with no central alleles (13 females and 8 males were new adults in 2012). In spring 2013, we captured 22 female/13 male hybrids (with central alleles), with 17 females and 12 males as new hybrid adults; these adult hybrids could have been F1's, progeny of F1 x F1's or F1 backcrosses. We also captured 25 females/8 males with no central alleles, with 12 females and 6 males that were new adults in spring 2013. Using male population size estimates for 2011 (see above), and the number of translocated males that survived greater than three days ($n = 5$), we calculated the expected number of F1 hybrids in spring 2012. Similarly, using the observed number of F1 hybrid adults and individuals that did not carry central alleles in 2012, we calculated the expected number of new hybrid adults (e.g., carrying central alleles) in spring 2013. We used a $\chi^2$ test to determine differences between the observed and expected number of hybrid individuals in both years. We bootstrapped the assigned number of genotypes to hybrids or non-hybrids and then computed the relative fitness of the bootstrapped values relative to expectations 1000 times to get the confidence intervals. A Fisher's exact test was used to determine differences between the number of hybrid and non-hybrid females that carried a full complement (4) of pouch young in spring 2012 (hybrids females with/without four pouch young = 8/0, non-hybrid females with/without four pouch young = 13/8). We also compared weight, head, body and tail length (dependent variables) for hybrid and non-hybrid (genetic background; independent variable) new adult females/males in spring 2012 (hybrid females/males n = 8/5, non-hybrid females/males n = 12/8) and weight, head, body and tail length for hybrid/non-hybrid new adults in 2013 (hybrid females/males n = 16/12, non-hybrid females/males n = 11/5) using ANOVAs (SPSS ver 22, IBM, St Leonards, NSW, Australia; all data were tested for normality/equal variances). We examined survival for 2012 new adults including trapping results up until spring 2015. A Fisher's exact test was used to test for a difference in the number of F1 hybrids/non-hybrids still alive in spring 2015 compared with that expected from initial numbers detected in spring 2012.

**Code availability**. R scripts for the mark-recapture analyses are available in the Supplementary Information file as Supplementary Methods.

**Data availability**. New microsatellite loci sequences are available in GenBank (accession numbers MF568684-MF568699). The genotypic data that support the findings of this study are available from the Dryad Digital Repository (http://dx.doi.org/10.5061/dryad.7g988)[36]. All the other data that support the findings of this study are included in the manuscript and supplementary files or are available from the corresponding author (A.R.W.) upon reasonable request.

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

## Acknowledgements

We thank Rudi Pleschutschnig, Alison Kirkwood, Vinnie Antony, David McCoombe, Anthony Bock, Josh Griffiths and Paul Mitrovski for help with field work, Rupert Baker for veterinary support, Tom Pelly, Georgina Boardman and Jerry Alexander for logistical support, and Carla Sgrò for comments on the manuscript. We also thank the Mt Buller Mt Stirling Resort Management Board for ongoing support of this project. Financial support was provided by the Mt Buller Mt Stirling Resort Management Board, FAME Ltd, the Department of Sustainability and Environment (DSE) Victoria, and the Department of Land, Water and Planning (DELWP) Victoria (Hume Region). A.R.W. and A.A.H. also received financial support from the National Environment Science Program Threatened Species Recovery Hub (Federal Department of Environment and Energy) and the Australian Research Council Discovery grant scheme (DP160100661). Field work permits were provided by responsible state agencies (Animal Ethics: DSE ARI 10/16, DPI WSI 25.12, DEPI ARI 14/10; Wildlife Research Permits: 10004130, 10005612, 10006441, 10007208).

## Author contributions

A.R.W., D.H., L.P. and I.M.: Conceptualised the project. D.H.: Undertook most of the fieldwork, with help from T.K., A.R.W., L.P., and I.M. Genotyping was undertaken by A.v.R. and A.R.W, while mark-recapture analyses were undertaken by J.S. All genetic and phenotypic analyses were undertaken by A.R.W. and A.A.H. The manuscript was written by A.R.W. and A.A.H., with contributions from J.S., D.H., L.P. and I.M. All authors reviewed the paper prior to submission.

## Additional information

**Competing interests:** The authors declare no competing financial interests.

