## [Peer Review File · Nature Communications]

Reviewers' comments:

Reviewer #1 (Remarks to the Author):

This manuscript describes a genetic rescue attempt in the endangered Mountain pygmy possum in Australia. There have been a modest number of genetic rescues of threatened species described in the literature (e.g. the Florida panther, the Mexican wolf), so it is instructive to ask how this study differs. I note the following:

1. To my knowledge this is the first documented case of genetic rescue of a marsupial in a natural wild environment
2. The populations being combined have been isolated for $\sim 20,000$ years, likely the longest isolation where genetic rescue at human hands has been attempted for conservation purposes
3. The study involves some F2 individuals so there is some information on the persistence of beneficial fitness effects.
4. The work has been done with praiseworthy rigour in experimental design, experimental methodology and statistical analyses, with care being taken to exclude non-genetic factors as the cause of the rescue observed following gene flow.

This case should be of interest to many readers.

The authors represent an extremely strong team in evolutionary and conservation genetics.

The presentation is good, but I did find it a little heavy going in places.

I disagree with the author's recommendation that genetic rescue be considered a viable option only when there are large healthy populations available as a genetic. Using outbred immigrants is certainly a superior option to using inbred ones, but inbred immigrants are expected to yield about 2/3 the benefits of outbred ones in the long term (F3 and later generations) for equally inbred target populations (Frankham 2015). In line with this, empirical evidence yielded a median genetic rescue benefit of 52% across 87 comparisons using inbred immigrants, and this was approximately 1/2 the median benefits for outbred immigrants. For example, Heber et al. (2013) reported substantial benefits in survival, recruitment, proportion of normal sperm and immune response for rescues using inbred immigrants in the South Island robin in New Zealand.

For species that have declined to only two isolated inbred populations (or only two such populations with a low risk of outbreeding depression when crossed) this may be the only practical genetic option to save the species, and it is most unwise to preclude such an action. Would the authors have advised against the genetic rescue of Mexican wolves that was done by combining 3 small inbred populations? Would the authors advise against genetic rescue of the Western barred bandicoot that exists only as two inbred island populations in Western Australia?

Further, in the cross of two isolated inbred populations, most of the harmful recessive alleles will not be shared, so that natural selection can remove many of the harmful alleles over generations. Such selective advantages of immigrant alleles have been documented in several studies (e.g. Ebert et al. 2002; Saccheri & Brakefield 2002; Adams et al. 2011; Miller et al. 2012). While it can be argued that natural selection will be weakened in small populations, it will not be eliminated. For example, there was a substantial change over generations in the frequency of genetic material from an immigrant in the small Isle Royale gray wolf population (Adams et al. 2011).

Finally, heterozygosity (0.552) in the Mt Higginbotham population is low compared to non-threatened species and populations so there are grounds for concluding that the immigrants used in this study are mildly inbred. For example, heterozygosity in *Burramys parvus* ranks with threatened and bottlenecked marsupial population in the review of Eldridge et al. (2010), rather than non-threatened ones (heterozygosity range 0.6-0.89, with most values in the 0.7-0.8 range), and is lower than all non-threatened populations. The authors do not explicitly assess the risk of outbreeding depression in their crossed population. The populations are likely adapted to similar environmental conditions, so it is unlikely for this reason. There is no mention of an evaluation of whether the chromosomes show fixed difference, so it is not possible to eliminate that as a risk factor, especially as marsupials generally show high rates of chromosomal evolution. However, this may not apply to mountain pygmy possums (Westerman et al. 2010). The 20,000 years of isolation does represent an elevated risk of outbreeding depression.

Detailed comments:

Line 8: 'CESAR' for 'cesar'

Lines 23-24: 'but did not differ in lifespan' – this is contradicted for females by lines 99-101.

Lines 115-121: I had to read this material a few times to understand it. It would be improved by modifying the 3rd sentence to say 'The incidence of F2s (12.8%) and backcrossed individuals (48.9%) observed in the population were non-significantly higher than those expected.' The observed numbers are 49% and 17.5% greater than those expected. In this and a number of other cases, the non-significant differences are likely a consequence of low statistical power (and most of the trends seem to favour hybrid animals), a point that might well be mentioned.

Line 349: Italicise '*Burramys parvus*'

Line 354: 'Garcia-Dorado' for 'Carcia-Dorado'

p.17 Figure 1: The authors might consider adding arrows to indicate the introduction of immigrants that had an opportunity to breed, i.e. in 2011 and 2014.

p.18 Figure 2: Please add a key to distinguish the results for non-hybrid and hybrid progeny.

References

- Adams, J.R., Vucetich, L.M., Hedrick, P.W., et al., 2011. Genomic sweep and potential genetic rescue during limiting environmental conditions in an isolated wolf population. *Proceedings of the Royal Society B: Biological Sciences* 278, 3336-3344.
- Ebert, D., Haag, C., Kirkpatrick, M., et al., 2002. A selective advantage to immigrant genes in a *Daphnia* metapopulation. *Science* 295, 485-488.
- Eldridge, M.D., Piggott, M., Hazlitt, S.L., 2010. Population genetic studies of the Macropodoidea: a review, In *Macropods: The Biology of Kangaroos, Wallabies and Rat-Kangaroos*. CSIRO Publishing, Melbourne. pp. 35-51.
- Frankham, R., 2015. Genetic rescue of small inbred populations: meta-analysis reveals large and consistent benefits of gene flow. *Molecular Ecology* 24, 2610-2618.
- Heber, S., Varsani, A., Kuhn, S., et al., 2013. The genetic rescue of two bottlenecked South Island robin populations using translocations of inbred donors. *Proceedings of the Royal Society B-Biological Sciences* 280.

Miller, J., Poissant, J., Hogg, J., et al., 2012. Genomic consequences of genetic rescue in an insular population of bighorn sheep (*Ovis canadensis*). *Molecular Ecology* 21, 1583-1596.

Saccheri, I.J., Brakefield, P.M., 2002. Rapid spread of immigrant genomes into inbred populations. *Proceedings of the Royal Society of London B: Biological Sciences* 269, 1073-1078.

Westerman, M., Meredith, R.W., Springer, M.S., 2010. Cytogenetics meets phylogenetics: A review of karyotype evolution in Diprotodontian marsupials. *Journal of Heredity* 101, 690-702.

Dick Frankham

Reviewer #2 (Remarks to the Author):

This manuscript addresses a very important question in conservation biology: What are the genetic costs and benefits of genetic rescue attempts? Identifying these costs and benefits is not easy, because it requires long-term monitoring in the field and the separation of genetic from environmental effects. Both are challenging and hence there are very few reliable estimates available. Thus, estimates of these costs and benefits are very important and this manuscript attempts to do so in a critically endangered marsupial.

As it stands, I think the MS suffers from two major drawbacks that would need to be resolved before meaningful estimates of the costs and benefits of genetic rescue in this species can be obtained.

First, the authors claim that they can separate the genetic benefits of translocations from the effects of an environmental improvement program, because the environmental improvement was carried out in 2008-2011 but the translocations only started in 2012 (lines 58-63). To me this is not a compelling argument. The pattern shown in Fig.1 is compatible with a scenario where removal of an environmental factor that kept the population at very low numbers, e.g. introduced predators, leads to exponential population growth. Thus, the strong population growth in the years following the translocations could simply be due to the exponential growth that resulted from the environmental improvement program in the previous years. To convincingly argue that the population growth post-2012 is due to genetic rescue rather than environmental improvement one would have to explicitly account for the environmental improvements and then show that the genetic rescue effects contributed to population growth above and beyond the (potentially lagged) benefits of environmental improvements.

Second, the estimates of the potential costs of genetic rescue (outbreeding depression) are based on the assumption of random mating of hybrid individuals. Although this is not stated explicitly, the calculations of the expected proportions of F₂s and backcrosses assume random mating and all the subsequent conclusions about survival are only correct given random mating. If mating among hybrids and non-hybrids is non-random, then the proportion of F₂s in the population contains little information about their survival. This needs to be addressed in order to say anything about the strength of outbreeding

depression.

Detailed comments:

lines 37-38: It may be worth repeating here that most translocations are accompanied by environmental manipulations, making it difficult to separate genetic from environmental effects.

line 54: Please indicate on Extended Data Figure 2 where the Federation-Wombat bowl is located.

line 51: You use three different terms for these artificial boulderfields: 'reconstituting' here, 're-created' in the caption to Extended Data Fig. 2, and 'created' in the legend to Extended Data Fig. 2. Please standardize.

line 74: Levels of heterozygosity have certainly increased markedly, but they still differ from those at Mt Higginbotham (expected heterozygosity 0.416 versus 0.552). Thus, the effective population size at Mt Buller is still smaller than at Mt Higginbotham. That Mt Buller does not yet have levels of genetic diversity comparable to the other populations is further supported by the fact that they have, on average, one allele less per locus.

lines 83-86: I struggled with the change of focus in these sentences. The first sentence presents the observed fitness differences as heterosis (although it doesn't call it that), the second as inbreeding depression. This is a little confusing for the reader. I would stick to heterosis here since the paper is about genetic rescue, not inbreeding depression.

lines 98-101: I struggled with the presentation of the data on longevity here. Firstly, there was no explanation of what μ represents and in what units it was measured, making interpretation of the survival data impossible. Secondly, the first part of the sentence (no difference in survival) directly contradicts the second (significant difference in the proportion alive). These statistical analyses need to be explained in a lot more detail for a reader to comprehend.

lines 120-121: See my general comment above. In addition, I cannot see how one can say anything about the mating of the F2s and backcrosses. The data report the proportion of F2s and backcrosses in the population but nothing about their mating or mating success. Hence, I cannot see how one can come to conclusions about the mating of F2s and backcrosses.

lines 143-144: This advice may perhaps be a little simplistic. Large healthy populations have lower expected drift load but larger segregating mutational load. Thus, the probability of introducing mildly deleterious mutations is indeed lower from a large source population, but the likelihood of introducing a major deleterious mutation is higher. Thus, without

further detailed explanations as to why on balance larger source populations should be preferred, the advice given here is difficult to understand.

line 196: It would be helpful to know why only males were translocated.

lines 233-234: Why were translocated individuals omitted from the analyses? They are part of the population, hence my naive expectation would be that they should be part of a population size estimate.

lines 245-246: Extended Data Table 1 is rather cryptic (see below) but if I deciphered it correctly you did not model any of the parameters in a sex-specific way. However, you report (lines 96-97) that females survive longer than males. Should this heterogeneity not be accounted for in the capture-recapture models? A similar argument would apply to the apparent differences in recapture rates (line 249).

line 247: Perhaps I missed it but which model was the final one?

line 248: I failed to understand why you need an estimate of male population size to obtain an estimate of male gene frequencies. Normally, one would calculate the male gene frequency by using all the sampled males, not the male population size. I cannot see how having an estimate of male population size corrected for undetected individuals would help, since one will not have the genetic data for such individuals.

lines 269-270: It was not obvious to me that the independent allele frequency model was the best choice here. Please justify.

line 273: The generation time is typically shorter than the average longevity. Why did the authors use average lifespan as a measure of generation time?

Figure 2: Perhaps I overlooked it but I could not find a key as to what the grey and blue colors signify.

Extended Data:

Fig.1: Please specify the units in which altitude is measured. Also, what is the black line that crosses the map? A river or a road?

Fig.2: Please specify in the legend what A, B, C, and D stand for.

Line 416: This should be Figure 3 not Figure 2.

Fig. 3: Please provide more help to the reader in the caption. What does this figure show? I couldn't see how this figure shows 'introduced alleles within individuals'. What is on the x-axis, what on the y-axis?

Table 1: Several of the parameters remained cryptic even after reading the methods in detail. γ' and γ'' are temporary emigration parameters but what do they refer to? And why are there two different parameters? π is a class membership probability, but what classes is it referring to? Also, why do the full model and the model just below it in the table have the same number of parameters? The second to last model ought to have fewer parameters given that both s and π are invariant across primary occasions. Please explain in much more detail what is happening here. As it stands, I would have no chance to repeat the analyses presented here.

Reviewer #3 (Remarks to the Author):

The authors report a successful attempt to boost a highly endangered population of pygmy possums by translocating conspecifics over multiple years. The authors claim the results provide evidence that genetic rescue from a large, well-adapted population into a small, inbred population has boosted the latter, as shown by increases in genetic variation, population size, size of individuals, and the survival of hybrid offspring. A fundamental problem with the study is that it is limited to observation of a single population over time, without any untreated (control) populations. So, contrary to the authors' assertion, it is not possible to truly "separate the importance of genetic versus ecological factors in the recovery process". Further, the effects of genetic versus demographic rescue cannot be disentangled because immigrant individuals were added without removal of resident individuals. Immigrants provided both new genetic variation and boosted the population size. Advocating for "large healthy populations" as sources of immigrants is an assertion unsupported by any data in this paper because it was not tested by comparing the effects of immigrants from large and small populations on replicate recipient populations. This study is similar to a few previous ones of wolves 1, adders 2, bighorn sheep 3, and Florida panthers 4 where there is evidence for immigrants relieving inbreeding depression, but genetic rescue could be definitively identified because of a lack of replication at the population level and confounding of demographic and genetic rescue effects. The present paper does a solid job of exploring changes in genetic variation, population size, individual size, and identifying the relative contributions to population growth of hybrid and non-hybrid individuals. Clearly, solid field and lab work have gone into this report. The statistical analyses are sound. Unfortunately, the constraint of having only a single study population makes it impossible to rule out factors that confound the interpretation of genetic rescue as responsible for the growth of this population.

In addition to the papers cited above, the present manuscript might also gain context from a recent meta-analysis of genetic rescue 5.

1 Adams, J.R., Vucetich, L.M., Hedrick, P.W., Peterson, R.O., & Vucetich, J.A., Genomic sweep and potential genetic rescue during limiting environmental conditions in an isolated wolf population. *Proceedings of the Royal Society of London, Series B* 278 (1723), 3336-3344 (2011).

2 Madsen, T., Shine, R., Olsson, M., & Wittzell, H., Conservation biology: Restoration of an inbred adder population. *Nature* 402, 34-35 (1999).

3 Hogg, J.T., Forbes, S.H., Steele, B.M., & Luikart, G., Genetic rescue of an insular population of large mammals. *Proceedings of the Royal Society of London, Series B* 273 (1593), 1491-1499 (2006).

4 Johnson, W.E. et al., Genetic restoration of the Florida panther. *Science* 329, 1641-1645 (2010).

5 Frankham, R., Genetic rescue of small inbred populations: meta-analysis reveals large and consistent benefits of gene flow. *Molecular Ecology* 24 (11), 2610-2618 (2015).

Responses to reviewers' comments:

Response to Reviewer #1:

This manuscript describes a genetic rescue attempt in the endangered Mountain pygmy possum in Australia. There have been a modest number of genetic rescues of threatened species described in the literature (e.g. the Florida panther, the Mexican wolf), so it is instructive to ask how this study differs. I note the following:

- 1. To my knowledge this is the first documented case of genetic rescue of a marsupial in a natural wild environment*
- 2. The populations being combined have been isolated for ~ 20,000 years, likely the longest isolation where genetic rescue at human hands has been attempted for conservation purposes*
- 3. The study involves some F2 individuals so there is some information on the persistence of beneficial fitness effects.*
- 4. The work has been done with praiseworthy rigour in experimental design, experimental methodology and statistical analyses, with care being taken to exclude non-genetic factors as the cause of the rescue observed following gene flow.*

This case should be of interest to many readers.

The authors represent an extremely strong team in evolutionary and conservation genetics. The presentation is good, but I did find it a little heavy going in places.

Response: We thank the reviewer for the positive comments and recognising the innovative aspects of our work.

I disagree with the author's recommendation that genetic rescue be considered a viable option only when there are large healthy populations available as a genetic. Using outbred immigrants is certainly a superior option to using inbred ones, but inbred immigrants are expected to yield about 2/3 the benefits of outbred ones in the long term (F3 and later generations) for equally inbred target populations (Frankham 2015). In line with this, empirical evidence yielded a median genetic rescue benefit of 52% across 87 comparisons using inbred immigrants, and this was approximately 1/2 the median benefits for outbred immigrants. For example, Heber et al. (2013) reported substantial benefits in survival, recruitment, proportion of normal sperm and immune response for rescues using inbred immigrants in the South Island robin in New Zealand.

For species that have declined to only two isolated inbred populations (or only two such populations with a low risk of outbreeding depression when crossed) this may be the only practical genetic option to save the species, and it is most unwise to preclude such an action. Would the authors have advised against the genetic rescue of Mexican wolves that was done by combining 3 small inbred populations? Would the authors advise against genetic rescue of the Western barred bandicoot that exists only as two inbred island populations in Western Australia? Further, in the cross of two isolated inbred populations, most of the harmful recessive alleles will not be shared, so that natural selection can remove many of the harmful alleles over generations. Such selective advantages of immigrant alleles have been documented in several studies (e.g. Ebert et al. 2002; Saccheri & Brakefield 2002; Adams et al. 2011; Miller et al. 2012). While it can be argued that natural selection will be

weakened in small populations, it will not be eliminated. For example, there was a substantial change over generations in the frequency of genetic material from an immigrant in the small Isle Royale gray wolf population (Adams et al. 2011).

Response: We did not mean to preclude this option when only small populations are available, and we appreciate there can be benefits. Others (eg. Hedrick) have pointed out that there are issues associated with this approach despite short term advantages, but given that this is not the main concern of the current paper and not tested here, we have removed throughout the manuscript any recommendation that genetic rescue only be considered when a large healthy population(s) is available. Similarly, we have included in the discussion a statement about the possibility of crossing inbred populations.

*Finally, heterozygosity (0.552) in the Mt Higginbotham population is low compared to non-threatened species and populations so there are grounds for concluding that the immigrants used in this study are mildly inbred. For example, heterozygosity in *Burramys parvus* ranks with threatened and bottlenecked marsupial population in the revue of Eldridge et al. (2010), rather than non-threatened ones (heterozygosity range 0.6-0.89, with most values in the 0.7-0.8 range), and is lower than all non-threatened populations.*

Response: Our initial work on this species (Mitrovski et al. 2007) showed heterozygosity in the central region to range between 0.6 and 0.69 (Mt Higginbotham was 0.67) using 8 microsatellite markers, which puts it into the “non-threatened” range. We designed 16 new microsatellite markers specifically for the Mt Buller population and this study (e.g. selecting a range of markers that included some with fixed/non-overlapping allelic differences between Mt Buller and Mt Higginbotham) and therefore this tended to reduce heterozygosity. If we only use the original 8 microsatellite loci for the Mt Higginbotham 2012 data, then $H_E = 0.67$ (the same as in Mitrovski et al. 2007). We have added a sentence to the Methods mentioning that some loci were chosen because they had fixed and/or non-overlapping allelic differences between Mt Buller and the central region so that we could track hybrids.

The authors do not explicitly assess the risk of outbreeding depression in their crossed population. The populations are likely adapted to similar environmental conditions, so it is unlikely for this reason. There is no mention of an evaluation of whether the chromosomes show fixed difference, so it is not possible to eliminate that as a risk factor, especially as marsupials generally show high rates of chromosomal evolution. However, this may not apply to mountain pygmy possums (Westerman et al. 2010). The 20,000 years of isolation does represent an elevated risk of outbreeding depression.

Response: We have now added several sentences discussing the risk of outbreeding depression in the Discussion. We assumed the risk to be low due to the similar environments and the likely impact of drift on the Mt Buller population (see Weeks et al. 2016).

Detailed comments:

Line 8: 'CESAR' for 'cesar'

Response: cesar is a company name and all in lower case.

Lines 23-24: 'but did not differ in lifespan' – this is contradicted for females by lines 99-101.

Response: changed.

Lines 115-121: I had to read this material a few times to understand it. It would be improved by modifying the 3rd sentence to say 'The incidence of F2s (12.8%) and backcrossed individuals (48.9%) observed in the population were non-significantly higher than those expected.' The observed numbers are 49% and 17.5% greater than those expected. In this and a number of other cases, the non-significant differences are likely a consequence of low statistical power (and most of the trends seem to favour hybrid animals), a point that might well be mentioned.

Response: We have changed the wording as suggested.

Line 349: Italicise 'Burrmys parvus'

Response: Changed

Line 354: 'Garcia-Dorado' for 'Carcia-Dorado'

Response: Changed

p.17 Figure 1: The authors might consider adding arrows to indicate the introduction of immigrants that had an opportunity to breed, i.e. in 2011 and 2014.

Response: We have added arrows to Figure 1 to indicate when we introduced males in 2011 and 2014.

p.18 Figure 2: Please add a key to distinguish the results for non-hybrid and hybrid progeny.

Response: We have added text to the Figure 2 legend to indicate hybrids and non-hybrids.

References

Adams, J.R., Vucetich, L.M., Hedrick, P.W., et al., 2011. Genomic sweep and potential genetic rescue during limiting environmental conditions in an isolated wolf population. Proceedings of the Royal Society B: Biological Sciences 278, 3336-3344.

Ebert, D., Haag, C., Kirkpatrick, M., et al., 2002. A selective advantage to immigrant genes in a Daphnia metapopulation. Science 295, 485-488.

Eldridge, M.D., Piggott, M., Hazlitt, S.L., 2010. Population genetic studies of the Macropodoidea: a review, In Macropods: The Biology of Kangaroos, Wallabies and Rat-Kangaroos. CSIRO Publishing, Melbourne. pp. 35-51.

Frankham, R., 2015. Genetic rescue of small inbred populations: meta-analysis reveals large and consistent benefits of gene flow. Molecular Ecology 24, 2610-2618.

Heber, S., Varsani, A., Kuhn, S., et al., 2013. The genetic rescue of two bottlenecked South Island robin populations using translocations of inbred donors. Proceedings of the Royal Society B-Biological Sciences 280.

*Miller, J., Poissant, J., Hogg, J., et al., 2012. Genomic consequences of genetic rescue in an insular population of bighorn sheep (*Ovis canadensis*). Molecular Ecology 21, 1583-1596.*

Saccheri, I.J., Brakefield, P.M., 2002. Rapid spread of immigrant genomes into inbred populations. Proceedings of the Royal Society of London B: Biological Sciences 269, 1073-1078.

Westerman, M., Meredith, R.W., Springer, M.S., 2010. Cytogenetics meets phylogenetics: A review of karyotype evolution in Diprotodontian marsupials. Journal of Heredity 101, 690-702.

Response: We have now included some of these references in the manuscript where appropriate.

Response to Reviewer #2:

This manuscript addresses a very important question in conservation biology: What are the genetic costs and benefits of genetic rescue attempts? Identifying these costs and benefits is not easy, because it requires long-term monitoring in the field and the separation of genetic from environmental effects. Both are challenging and hence there are very few reliable estimates available. Thus, estimates of these costs and benefits are very important and this manuscript attempts to do so in a critically endangered marsupial.

As it stands, I think the MS suffers from two major drawbacks that would need to be resolved before meaningful estimates of the costs and benefits of genetic rescue in this species can be obtained.

First, the authors claim that they can separate the genetic benefits of translocations from the effects of an environmental improvement program, because the environmental improvement was carried out in 2008-2011 but the translocations only started in 2012 (lines 58-63). To me this is not a compelling argument. The pattern shown in Fig.1 is compatible with a scenario where removal of an environmental factor that kept the population at very low numbers, e.g. introduced predators, leads to exponential population growth. Thus, the strong population growth in the years following the translocations could simply be due to the exponential growth that resulted from the environmental improvement program in the previous years. To convincingly argue that the population growth post-2012 is due to genetic rescue rather than environmental improvement one would have to explicitly account for the environmental improvements and then show that the genetic rescue effects contributed to population growth above and beyond the (potentially lagged) benefits of environmental improvements.

Response: It would of course be great to have populations from similar areas that one could use in a direct comparison but that is perhaps never likely to be the case when we are dealing with threatened species. Nevertheless, we have now included population size data in the Supplementary Information (and referenced in the main text) from two populations at a ski resort in the central region where both predator control and habitat improvements

were implemented at the same time as at Mount Buller. Despite these environmental improvements, there was no increase in population size at either site as we saw at Mount Buller. This at least provides additional supporting data that supports our conjecture. However we agree with the reviewer that we have not been able to directly separate the genetic and ecological effects in the Mount Buller population – and we have now amended the manuscript and chosen our words more carefully in this regard. Nevertheless, we have clearly shown that hybrids had a substantial fitness benefit that has resulted in an increase in population size – which suggests that the pattern in Fig 1 is not simply due to environmental improvements. Similarly, the new population size data included in the Supplementary Information also suggest that environmental improvements are unlikely to explain the large increase in pop size at Mount Buller.

Second, the estimates of the potential costs of genetic rescue (outbreeding depression) are based on the assumption of random mating of hybrid individuals. Although this is not stated explicitly, the calculations of the expected proportions of F2s and backcrosses assume random mating and all the subsequent conclusions about survival are only correct given random mating. If mating among hybrids and non-hybrids is non-random, then the proportion of F2s in the population contains little information about their survival. This needs to be addressed in order to say anything about the strength of outbreeding depression.

Response: We have now added the caveat that we are assuming that hybrids mate randomly within the population. Data from captive populations provide no evidence for non-random mating and this is now mentioned, but we agree that random mating in the field remains an assumption.

Detailed comments:

lines 37-38: It may be worth repeating here that most translocations are accompanied by environmental manipulations, making it difficult to separate genetic from environmental effects.

Response: We have included a line to this effect.

line 54: Please indicate on Extended Data Figure 2 where the Federation-Wombat bowl is located.

Response: Now indicated in the Figure legend.

line 51: You use three different terms for these artificial boulderfields: 'reconstituting' here, 're-created' in the caption to Extended Data Fig. 2, and 'created' in the legend to Extended Data Fig. 2. Please standardize.

Response: Changed to 'created'.

line 74: Levels of heterozygosity have certainly increased markedly, but they still differ from those at Mt Higginbotham (expected heterozygosity 0.416 versus 0.552). Thus, the effective

population size at Mt Buller is still smaller than at Mt Higginbotham. That Mt Buller does not yet have levels of genetic diversity comparable to the other populations is further supported by the fact that they have, on average, one allele less per locus.

Response: Have added the word 'approaching' given that we expect heterozygosity to still increase further since the 2014 translocation.

lines 83-86: I struggled with the change of focus in these sentences. The first sentence presents the observed fitness differences as heterosis (although it doesn't call it that), the second as inbreeding depression. This is a little confusing for the reader. I would stick to heterosis here since the paper is about genetic rescue, not inbreeding depression.

Response: We are talking about genetic rescue by heterosis (hybrid vigour); or to put another way, restoration of population fitness through the alleviation of inbreeding depression. While we have not tested for inbreeding depression, the second sentence merely provides a reason (intense inbreeding) for why there is likely to be inbreeding depression, and therefore why we saw a heterotic effect. We have added the words 'heterotic effect' to the first sentence, but left the second sentence as is.

lines 98-101: I struggled with the presentation of the data on longevity here. Firstly, there was no explanation of what μ represents and in what units it was measured, making interpretation of the survival data impossible. Secondly, the first part of the sentence (no difference in survival) directly contradicts the second (significant difference in the proportion alive). These statistical analyses need to be explained in a lot more detail for a reader to comprehend.

Response: μ is the mean survival in years. We have now clarified this. The first test was a MANOVA (mentioned in Methods) comparing hybrid and non-hybrid female survival. While hybrid females tended to survive longer, this was non-significant. However, there were still hybrid F1 females alive in spring 2015, whilst no non-hybrid females were alive – this was significantly different from what was expected (based on a Fisher's exact test). We have now mentioned this second test in the Methods as well.

lines 120-121: See my general comment above. In addition, I cannot see how one can say anything about the mating of the F2s and backcrosses. The data report the proportion of F2s and backcrosses in the population but nothing about their mating or mating success. Hence, I cannot see how one can come to conclusions about the mating of F2s and backcrosses.

Response: We have added a missing word in this sentence (F1s).

lines 143-144: This advice may perhaps be a little simplistic. Large healthy populations have lower expected drift load but larger segregating mutational load. Thus, the probability of introducing mildly deleterious mutations is indeed lower from a large source population, but the likelihood of introducing a major deleterious mutation is higher. Thus, without further detailed explanations as to why on balance larger source populations should be preferred, the advice given here is difficult to understand.

Response: We have removed/changed this statement as per Reviewer 1's comments above.

line 196: It would be helpful to know why only males were translocated.

Response: We have added this to the Methods.

lines 233-234: Why were translocated individuals omitted from the analyses? They are part of the population, hence my naive expectation would be that they should be part of a population size estimate.

Response: Translocated individuals artificially inflate figures in the year of the translocation, hence this was why they were removed from the population size analyses (from translocated males, only a single male persisted for multiple years). We therefore only included individuals that were born on the mountain in population estimates. We have now made this clear in the Methods.

lines 245-246: Extended Data Table 1 is rather cryptic (see below) but if I deciphered it correctly you did not model any of the parameters in a sex-specific way. However, you report (lines 96-97) that females survive longer than males. Should this heterogeneity not be accounted for in the capture-recapture models? A similar argument would apply to the apparent differences in recapture rates (line 249).

Response: The reviewer is correct. We have now added a gender covariate in the analysis to account for sex in survival and capture probabilities, that is, we now model both survival and capture probabilities as functions of the sex covariate. Once again we used the BIC for model selection. The inclusion of the sex covariate improved the model fit but population size estimates were very similar to our previous results. As expected, the gender effect was significant for both possum survival and capture rates, again indicating that females have greater survival and capture rates compared with males.

line 247: Perhaps I missed it but which model was the final one?

Response: The final model was the model selected with the smallest BIC. This was Model 8 (as now mentioned in the Methods). This selected model is described as follows: survival probabilities were constant across primary occasions and dependent on gender, capture probabilities varied across primary occasions and were also dependent on gender, mixture probabilities varied across primary occasions; and temporary emigration was of Markovian type (see comment below for further details) and constant across primary occasions. We now describe the final chosen model in the Methods.

line 248: I failed to understand why you need an estimate of male population size to obtain an estimate of male gene frequencies. Normally, one would calculate the male gene frequency by using all the sampled males, not the male population size. I cannot see how having an estimate of male population size corrected for undetected individuals would help, since one will not have the genetic data for such individuals.

Response: Apologies, this was not written clearly and we have now amended it. We needed an estimate of male population size so that we could calculate the expected number of hybrids in 2012 (based on the translocation of the Mount Higginbotham males into the population in 2011). We used an estimate, rather than the known number to be alive because we do not capture the entire population during trapping, and for this analysis it was important to have an estimate for the entire population.

lines 269-270: It was not obvious to me that the independent allele frequency model was the best choice here. Please justify.

Response: Independent allele frequencies model assumes that the allele frequencies in each population are likely to be independent – that is that they are reasonably different from each other (Pritchard et al. 2000, Genetics). In our case, we have two populations (males from the central region and resident Mount Buller individuals which are of the southern region) that have very different allele frequencies (Mitrovski et al. 2007, Mol Ecol) – hence this is why we chose the independent allele frequencies model with admixture (admixture is needed to identify hybrids). We've added this to the Methods.

line 273: The generation time is typically shorter than the average longevity. Why did the authors use average lifespan as a measure of generation time?

Response: Good point – we have now used generation time as calculated by survivorship probabilities and pouch young for females of different age classes. This comes out to 1.54 (average female longevity was 1.61 for the same period from 1996-2010).

Figure 2: Perhaps I overlooked it but I could not find a key as to what the grey and blue colors signify.

Response: We have now added a key to the Figure legend.

Extended Data:

Fig.1: Please specify the units in which altitude is measured. Also, what is the black line that crosses the map? A river or a road?

Response: Amended.

Fig.2: Please specify in the legend what A, B, C, and D stand for.

Response: We have now added this to the Figure legend.

Line 416: This should be Figure 3 not Figure 2.

Response: Changed.

Fig. 3: Please provide more help to the reader in the caption. What does this figure show? I

couldn't see how this figure shows 'introduced alleles within individuals'. What is on the x-axis, what on the y-axis?

Response: We have provided more information in the Figure legend to indicate what the figure shows.

Table 1: Several of the parameters remained cryptic even after reading the methods in detail. gamma' and gamma'' are temporary emigration parameters but what do they refer to? And why are there two different parameters? Pi is a class membership probability, but what classes is it referring to? Also, why do the full model and the model just below it in the table have the same number of parameters? The second to last model ought to have fewer parameters given that both s and pi are invariant across primary occasions. Please explain in much more detail what is happening here. As it stands, I would have no chance to repeat the analyses presented here.

Response: First, gamma' and gamma'' (that is, γ' and γ'') are natural parameters included in the robust design model to account for any temporary emigration that may be present in the population. There are two types of temporary emigration that can be modelled when using the robust design model, these are: the random type (*i.e.*, temporary emigration occurs randomly) or Markovian type (*i.e.*, an animal 'remembers' that it is off the study area from one occasion to the next). These two parameters allow us to set the assumed temporary emigration type, such that: when setting gamma'=gamma'' the temporary emigration is assumed to be random, and when setting gamma' and gamma'' as separate parameters the temporary emigration is assumed to be Markovian. Alternatively, we can remove temporary emigration altogether from the model. We considered both temporary emigration types in our analysis (and also included models without temporary emigration). We have included the above details in Supplementary Table 1. We found that (based on BIC) the temporary emigration was of Markovian type. Note however that we expected minimal temporary emigration across years (primary occasions).

π_j (π_j) is a class membership probability that arises when fitting finite mixture models within the robust design model. These parameters take latent (unobserved) structures, that is, capture probabilities are modelled as functions of latent variables. The purpose of using mixtures in the robust design model is to account for any unexplained heterogeneity between individuals. Any individual that is part of the population is assumed to belong to a class - the number and size of these classes is unknown and unobserved. The more classes one assumes in the model, the more unexplained heterogeneity one believes exists between individuals. The parameters π_j simply represent the probability an individual belongs to some class j . In our analysis, we used the default setting of two mixture classes, π_1 and π_2 , (this is the recommended number of classes to use when fitting mixture models in MARK; having too many classes may cause instability problems in the estimation procedure). We now give a similar explanation of the details above in the Methods section.

There was indeed a slight discrepancy in the number of reported model parameters for each model, thank you for noticing this. We investigated this further and found that (by default) MARK displays the number of 'estimated parameters' rather than the total number of parameters considered in the model (see Section 4-65 of the 'Program MARK' manual).

MARK reports estimated parameters because some model parameters are not estimable (this occurs when some parameters are unidentifiable or the likelihood is maximized near the boundary). For clarity, we now fit our models using the R-package *RMark* (we found *RMark* much easier to work with in the R environment). Note that *RMark* is built around and depends on MARK. Using *RMark* we extracted (and now report) the total number of parameters in the model.

We also developed a supplementary document so that our analysis can be reproduced in R. Further details on the models fitted and the overall population size estimation analysis is given in this supplementary document.

Response to Reviewer #3:

The authors report a successful attempt to boost a highly endangered population of pygmy possums by translocating conspecifics over multiple years. The authors claim the results provide evidence that genetic rescue from a large, well-adapted population into a small, inbred population has boosted the latter, as shown by increases in genetic variation, population size, size of individuals, and the survival of hybrid offspring. A fundamental problem with the study is that it is limited to observation of a single population over time, without any untreated (control) populations. So, contrary to the authors' assertion, it is not possible to truly "separate the importance of genetic versus ecological factors in the recovery process".

Response: See response to reviewer 1 above. We appreciate that we do not have definitive evidence and we have now included in Supplementary Information population size data (using the same robust models) from two sites within another ski resort (Mount Hotham) that also implemented environmental improvement programs (but no translocations) through habitat restoration and predator control at the same time as Mount Buller (see comments to reviewer 2). These are mentioned in the main paper. Despite these environmental improvements, nothing dramatically changed within these populations like what we saw at Mount Buller. While not perfect controls, they do demonstrate that environmental improvements do not necessarily translate into an increase in population size. We have been able to separate the fitness of hybrids and non-hybrids, demonstrating a clear fitness advantage – highlighting that genetic effects have contributed substantially to population growth. Granted, we haven't been able to directly separate out the "genetic" versus "ecological" effects – and we have now chosen our words more carefully throughout the manuscript to reflect this.

Further, the effects of genetic versus demographic rescue cannot be disentangled because immigrant individuals were added without removal of resident individuals. Immigrants provided both new genetic variation and boosted the population size.

Response: The number of immigrant individuals was small and 5 of the 6 translocated males only persisted for a single breeding season (the 6th male did not contribute until the 2013 breeding season). This does not constitute a large demographic boost to explain the increases in population size (please note that we also did not include any translocated individuals in the CMR population analyses). Moreover, population size did not increase in

2012 (the year after the males were added), as would be expected if it was purely a demographic boost. Our analyses indicate that the F1 progeny were more than twice as fit as the resident individuals (again highlighting a genetic reason for the increase in population size, rather than a “demographic rescue”). We have done some rewording to cover the above points.

Advocating for “large healthy populations” as sources of immigrants is an assertion unsupported by any data in this paper because it was not tested by comparing the effects of immigrants from large and small populations on replicate recipient populations.

Response: As per reviewer 1 (and 2) we have amended the manuscript to remove the assertion around “large healthy populations”.

This study is similar to a few previous ones of wolves 1, adders 2, bighorn sheep 3, and Florida panthers 4 where there is evidence for immigrants relieving inbreeding depression, but genetic rescue could be definitively identified because of a lack of replication at the population level and confounding of demographic and genetic rescue effects. The present paper does a solid job of exploring changes in genetic variation, population size, individual size, and identifying the relative contributions to population growth of hybrid and non-hybrid individuals. Clearly, solid field and lab work have gone into this report. The statistical analyses are sound. Unfortunately, the constraint of having only a single study population makes it impossible to rule out factors that confound the interpretation of genetic rescue as responsible for the growth of this population.

Response: We understand the reviewers point around replication, but replication of such a program for a threatened species in nature is often unrealistic as this experiment (along with the Florida panther/wolves/bighorn sheep and other examples) was born out of necessity to avoid extinction of the population! Regardless, as the reviewer points out, we clearly show strong fitness benefits associated with hybrids that have contributed to population growth – and this heterotic effect is a key requirement for demonstrating genetic rescue (see Tallmon et al. 2004 TREE; Whitely et al 2015 TREE).

In addition to the papers cited above, the present manuscript might also gain context from a recent meta-analysis of genetic rescue 5.

Response: We agree with the reviewer that this is an important reference in the context of our study and we had previously cited this important piece of work (ref number 3).

Reviewers' comments:

Reviewer #1 (Remarks to the Author):

The revision has satisfactorily addressed my queries

Reference 14 still has an error. The senior author's surname is Garcia-Dorado - it starts with a G, not a C.

Dick Frankham

Reviewer #2 (Remarks to the Author):

I enjoyed reading this revised and improved version of the manuscript. It tackles an important issue in conservation biology. However, as the first set of reviews and the answers of the authors have shown, the data set has some fairly substantial limitations: the effects of genetic rescue cannot be clearly separated from the effects of environmental improvements, and the calculations of outbreeding depression rely on a crucial assumption (random mating) that is unsubstantiated. Such limitations are common in data sets that are derived from conservation applications and do not by themselves provide a reason not to publish such work. However, given these limitations care should be taken to not oversell the results.

I feel that this manuscript is still overselling the results, and would profit from some more toning down. The caveats mentioned above mean that both the estimates of the benefits and the potential costs of genetic rescue are uncertain in this data set. This is ok, but it really needs to be reflected in the tone of the manuscript and the authors should refrain from black-and-white statements such as on lines 122-123, where the existence of outbreeding depression is completely negated. Even severe outbreeding depression will not remove all F2s or backcrosses, so the presence of F2 and backcrosses is not an argument against some outbreeding depression. The Results, the Discussion and the Abstract need to be worded more carefully to reflect the uncertainties in the results caused by the limitations of the data set.

On a more detailed note, I still struggle with the results of the survival analyses (lines 101-108), in part because the explanations in the Methods section (lines 353-359) are still too superficial. Statistical issues aside (for example, it does not become clear in the Methods what the different dependent and independent variables were in the MANOVA) I did not understand how mean lifespan of hybrid and non-hybrid F1 females could be 2.78 years and 1.8 years, respectively given that only data from 2012 and 2013 were used (lines 353-356). The first F1 offspring were detected in 2012 (line 322), when they would have been approximately half a year old. By 2013 they would have been 1.5 years old. How can the mean survival of hybrid females in this data set be 2.78 years, when the oldest they could have been in 2013 is approx. 1.5 years?

Some minor comments:

lines 95-98: Did I miss these data? I could not find them in the tables of figures.

line 99: remove the '(genetic load)'. Genetic load has been defined in many different ways (see for example the list in the famous textbook by Crow & Kimura, 1970) and without specifying which genetic load is meant here, this parenthetical mention of genetic load does not clarify things for the reader. The term 'deleterious alleles' is sufficient here.

line 166: It did not become clear to me what 'such a process' was referring to. The previous sentence refers to outbreeding depression, but this cannot be what the authors mean to refer to. Please clarify.

lines 331-333: Please provide a reference to the approach you chose to estimate generation time. As it stands, it would be impossible for a reader to replicate your calculations of generation time.

line 357: data are plural, hence was -> were

Table 2: How were the bootstrap confidence intervals estimates for hybrid relative fitness? I could not find a description in the Methods.

Reviewer #3 (Remarks to the Author):

The authors have done a good job of responding to the criticisms of the reviewers. The manuscript is improved in terms of diction and some statistical analyses. However, given the inherent limitations of study (lack of replication, inability to assign responses to treatment) some important criticisms cannot be addressed and prevent me from endorsing the manuscript. The study is useful, and adds to a growing list of similar studies, but it does not provide novel insights that will change thinking in the field. These are not criticisms of the authors, just constraints imposed by the study system.

Reviewers' comments:

Reviewer #1 (Remarks to the Author):

The revision has satisfactorily addressed my queries

Reference 14 still has an error. The senior author's surname is Garcia-Dorado - it starts with a G, not a C.

Dick Frankham

Reviewer #2 (Remarks to the Author):

I enjoyed reading this revised and improved version of the manuscript. It tackles an important issue in conservation biology. However, as the first set of reviews and the answers of the authors have shown, the data set has some fairly substantial limitations: the effects of genetic rescue cannot be clearly separated from the effects of environmental improvements, and the calculations of outbreeding depression rely on a crucial assumption (random mating) that is unsubstantiated. Such limitations are common in data sets that are derived from conservation applications and do not by themselves provide a reason not to publish such work. However, given these limitations care should be taken to not oversell the results.

I feel that this manuscript is still overselling the results, and would profit from some more toning down. The caveats mentioned above mean that both the estimates of the benefits and the potential costs of genetic rescue are uncertain in this data set. This is ok, but it really needs to be reflected in the tone of the manuscript and the authors should refrain from black-and-white statements such as on lines 122-123, where the existence of outbreeding depression is completely negated. Even severe outbreeding depression will not remove all F2s or backcrosses, so the presence of F2 and backcrosses is not an argument against some outbreeding depression. The Results, the Discussion and the Abstract need to be worded more carefully to reflect the uncertainties in the results caused by the limitations of the data set.

On a more detailed note, I still struggle with the results of the survival analyses (lines 101-108), in part because the explanations in the Methods section (lines 353-359) are still too superficial. Statistical issues aside (for example, it does not become clear in the Methods what the different dependent and independent variables were in the MANOVA) I did not understand how mean lifespan of hybrid and non-hybrid F1 females could be 2.78 years and 1.8 years, respectively given that only data from 2012 and 2013 were used (lines 353-356). The first F1 offspring were detected in 2012 (line 322), when they would have been approximately half a year old. By 2013 they would have been 1.5 years old. How can the mean survival of hybrid females in this data set be 2.78 years, when the oldest they could have been in 2013 is approx. 1.5 years?

Some minor comments:

lines 95-98: Did I miss these data? I could not find them in the tables of figures.

line 99: remove the '(genetic load)'. Genetic load has been defined in many different ways (see for example the list in the famous textbook by Crow & Kimura, 1970) and without specifying which genetic load is meant here, this parenthetical mention of genetic load does not clarify things for the reader. The term 'deleterious alleles' is sufficient here.

line 166: It did not become clear to me what 'such a process' was referring to. The previous sentence refers to outbreeding depression, but this cannot be what the authors mean to refer to. Please clarify.

lines 331-333: Please provide a reference to the approach you chose to estimate generation time. As it stands, it would be impossible for a reader to replicate your calculations of generation time.

line 357: data are plural, hence was -> were

Table 2: How were the bootstrap confidence intervals estimates for hybrid relative fitness? I could not find a description in the Methods.

Reviewer #3 (Remarks to the Author):

The authors have done a good job of responding to the criticisms of the reviewers. The manuscript is improved in terms of diction and some statistical analyses. However, given the inherent limitations of study (lack of replication, inability to assign responses to treatment) some important criticisms cannot be addressed and prevent me from endorsing the manuscript. The study is useful, and adds to a growing list of similar studies, but it does not provide novel insights that will change thinking in the field. These are not criticisms of the authors, just constraints imposed by the study system.

Response to reviewers' comments:

Response to Reviewer #1:

The revision has satisfactorily addressed my queries

Reference 14 still has an error. The senior author's surname is Garcia-Dorado - it starts with a G, not a C.

Dick Frankham

Response: We have made the adjustment to Ref 14.

Response to Reviewer #2:

I enjoyed reading this revised and improved version of the manuscript. It tackles an important issue in conservation biology. However, as the first set of reviews and the answers of the authors have shown, the data set has some fairly substantial limitations: the effects of genetic rescue cannot be clearly separated from the effects of environmental improvements, and the calculations of outbreeding depression rely on a crucial assumption (random mating) that is unsubstantiated. Such limitations are common in data sets that are derived from conservation applications and do not by themselves provide a reason not to publish such work. However, given these limitations care should be taken to not oversell the results.

I feel that this manuscript is still overselling the results, and would profit from some more toning down. The caveats mentioned above mean that both the estimates of the benefits and the potential costs of genetic rescue are uncertain in this data set. This is ok, but it really needs to be reflected in the tone of the manuscript and the authors should refrain from black-and-white statements such as on lines 122-123, where the existence of outbreeding depression is completely negated. Even severe outbreeding depression will not remove all F2s or backcrosses, so the presence of F2 and backcrosses is not an argument against some outbreeding depression. The Results, the Discussion and the Abstract need to be worded more carefully to reflect the uncertainties in the results caused by the limitations of the data set.

Response: We have deleted the sentence around outbreeding depression highlighted by the reviewer (lines 122-123) and added a sentence at the end of this paragraph being a lot more circumspect around evidence for a lack of outbreeding depression (lines 130-132). As suggested by the reviewer, we have also gone through the manuscript in detail and chosen our language more careful throughout, being careful not to oversell the results.

On a more detailed note, I still struggle with the results of the survival analyses (lines 101-108), in part because the explanations in the Methods section (lines 353-359) are still too superficial. Statistical issues aside (for example, it does not become clear in the Methods what the different dependent and independent variables were in the MANOVA) I did not understand how mean lifespan of hybrid and non-hybrid F1 females could be 2.78 years and 1.8 years, respectively given that only data from 2012 and 2013 were used (lines 353-356). The first F1 offspring were detected in 2012 (line 322), when they would have been approximately half a year old. By 2013 they would have been 1.5 years old. How can the mean survival of hybrid females in this data set be 2.78 years, when the oldest they could have been in 2013 is approx. 1.5 years?

Response: The dependent variables in the MANOVA were weight, head length, tail length, body length and survival, while the independent variable was genetic background (hybrid or non-hybrid). We have now indicated this in the Methods (lines 356-358). Mean lifespan was calculated for 2011/12 juvenile hybrid and non-hybrids up until last trapping in spring 2015 (which is a maximum of ~4 years old). We have now clarified this in the Methods (line 363).

Some minor comments:

lines 95-98: Did I miss these data? I could not find them in the tables of figures.

Response: We have added this to the Methods section (lines 355-356).

line 99: remove the '(genetic load)'. Genetic load has been defined in many different ways (see for example the list in the famous textbook by Crow & Kimura, 1970) and without specifying which genetic load is meant here, this parenthetical mention of genetic load does not clarify things for the reader. The term 'deleterious alleles' is sufficient here.

Response: We have removed '(genetic load)' from this sentence.

line 166: It did not become clear to me what 'such a process' was referring to. The previous sentence refers to outbreeding depression, but this cannot be what the authors mean to refer to. Please clarify.

Response: We mean genetic rescue, and we have now made this explicitly clear.

lines 331-333: Please provide a reference to the approach you chose to estimate generation time. As it stands, it would be impossible for a reader to replicate your calculations of generation time.

Response: Have added the reference (Grant & Grant 1992) to the Methods which provides details on the method we used.

line 357: data are plural, hence was -> were

Response: Changed

Table 2: How were the bootstrap confidence intervals estimates for hybrid relative fitness? I could not find a description in the Methods.

Response: We bootstrapped the assigned number of genotypes to hybrids or non-hybrids and then computed the relative fitness of the bootstrapped values relative to expectations 1000 times to get the confidence intervals. This is now mentioned (lines 351-353).

Response to Reviewer #3:

The authors have done a good job of responding to the criticisms of the reviewers. The manuscript is improved in terms of diction and some statistical analyses. However, given the inherent limitations of study (lack of replication, inability to assign responses to treatment) some important criticisms cannot be addressed and prevent me from endorsing the manuscript. The study is useful, and adds to a growing list of similar studies, but it does not provide novel insights that will change thinking in the field. These are not criticisms of the authors, just constraints imposed by the study system.

Response: We appreciate that there will also be a lack of replication and difficulties around interpretation in these types of studies on endangered species where genetic rescue (and habitat reconstruction) is critically important. We have therefore interpreted data

cautiously. However as outlined in the manuscript, the work provides evidence for genetic rescue over and above what has been possible in much of the previous work in this area, including the direct estimate of fitness and size of hybrids versus non hybrids and the ability to follow a population over an extended period.

REVIEWERS' COMMENTS:

Reviewer #2 (Remarks to the Author):

I appreciate the efforts the authors have made in revising the manuscript.

However, I understand even less now why a MANOVA was used to analyse differences in longevity. A MANOVA tests whether the vectors of means for two or more groups are different, and is usually employed when one wishes a single, overall test of a set of correlated dependent variables. Thus, the MANOVA employed by the authors tests whether hybrid status affected the vectors of means of survival, weight, head, body and tail length. Hence, the results reported on l. 117 are for the vectors above, and not for survival as stated in the MS. In addition, MANOVA cannot handle the fact that some of the animals are still alive and their longevity is thus unknown (right censored). For these reasons, I do not understand the choice of MANOVA here and the manuscript does not provide sufficient information as to why this statistical approach was chosen and how the results need to be interpreted.

Reviewers' comments:

Reviewer #2 (Remarks to the Author):

I appreciate the efforts the authors have made in revising the manuscript.

However, I understand even less now why a MANOVA was used to analyse differences in longevity. A MANOVA tests whether the vectors of means for two or more groups are different, and is usually employed when one wishes a single, overall test of a set of correlated dependent variables. Thus, the MANOVA employed by the authors tests whether hybrid status affected the vectors of means of survival, weight, head, body and tail length. Hence, the results reported on l. 117 are for the vectors above, and not for survival as stated in the MS. In addition, MANOVA cannot handle the fact that some of the animals are still alive and their longevity is thus unknown (right censored). For these reasons, I do not understand the choice of MANOVA here and the manuscript does not provide sufficient information as to why this statistical approach was chosen and how the results need to be interpreted.

Response to reviewers' comments:

Response to Reviewer #2:

I appreciate the efforts the authors have made in revising the manuscript.

However, I understand even less now why a MANOVA was used to analyse differences in longevity. A MANOVA tests whether the vectors of means for two or more groups are different, and is usually employed when one wishes a single, overall test of a set of correlated dependent variables. Thus, the MANOVA employed by the authors tests whether hybrid status affected the vectors of means of survival, weight, head, body and tail length. Hence, the results reported on l. 117 are for the vectors above, and not for survival as stated in the MS. In addition, MANOVA cannot handle the fact that some of the animals are still alive and their longevity is thus unknown (right censored). For these reasons, I do not understand the choice of MANOVA here and the manuscript does not provide sufficient information as to why this statistical approach was chosen and how the results need to be interpreted.

Response: We have now removed the MANOVA results from analyses and now only include the Fisher's exact test for showing a difference in survival of hybrids. This test does not suffer from the concerns the reviewer has raised about the MANOVAs. In addition, we have undertaken ANOVAs on the 2012 and 2013 cohort data for weight, head, body, and tail length for hybrid status and sex. This is now made clear in the Methods (Genetic and phenotypic analyses section) and Figure 2 legend.